# "There's something inside": Children's intuitions about animate agents

**Jonathan F. Kominsky** [1,2]*, **Patrick Shafto**[1], **Elizabeth Bonawitz**[1,2]

**1** Rutgers University–Newark, Chicago, NJ, United States of America, **2** Harvard University Graduate School of Education, Cambridge, England

* jonathan.f.kominsky@gmail.com

**Data Availability Statement:** All stimuli, data, and analysis files are available from our OSF repository at https://osf.io/3xzad/.

**Funding:** This project was supported in part by NSF SMA-1640816 to EB and PS, and by a

## Abstract

From infancy, humans have the ability to distinguish animate agents from inert objects, and preschoolers map biological and mechanical insides to their appropriate kinds. However, less is known about how identifying something as an animate agent shapes specific inferences about its internal properties. Here, we test whether preschool children (N = 92; North American population) have specifically biological expectations about animate agents, or if they have more general expectations that animate agents should have an internal source of motion. We presented preschoolers with videos of two puppets: a "self-propelled" fur-covered puppet, and a fur-covered puppet that is seen to be moved by a human actor. In addition, we presented preschoolers with images of a familiar artifact (motorcycle) and familiar animal (sheep). For each item, we asked them to choose what they thought was inside each of these entities: nothing, biological insides, or mechanical insides. Preschoolers were less likely to say that a self-propelled fur-covered object was empty, compared to a fur-covered object that was moved by a human actor, which converges with past work with infants. However, preschoolers showed no specifically biological expectations about these objects, despite being able to accurately match biological insides to familiar animals and mechanical insides to familiar artifacts on the follow-up measure. These results suggest that preschoolers do not have specifically biological expectations about animate agents as a category, but rather general expectations that such agents should not be empty inside.

## Introduction

As we navigate our environment one of the most critical distinctions we have to make among the entities we encounter are whether they are animate agents or inert objects. Animate agents can move unpredictably, and can be social partners, threats, or prey, and so we must be able to identify them quickly and from an early age [1]. However, merely distinguishing animate agents from inert objects is not, in and of itself, that useful. Rather, it is the information and intuitions that come along with the distinction that are critical for allowing us to navigate the world successfully. Merely identifying something as an animate agent is of little use without knowing something about the capabilities and properties of animate agents.

Research with infants has found that they are highly sensitive to the distinction between animate agents and inert objects [2]. In particular, infants seem to ascribe special importance

McDonnell Foundation Fellowship to EB. The funders had no role in study design, data collection and analysis, decision to publish, or preparation of the manuscript.

**Competing interests:** The authors have declared that no competing interests exist.

to whether an object engages in *self-propelled* motion, that is, motion with no apparent external source [3–5]. They make rich attributions to self-propelled objects, including ascribing psychological states like goals and intentions [6–9], and selectively make social evaluations about self-propelled entities [10,11].

In addition to attributing mental states and intentions to self-propelled objects, when these objects also have other features of real-world agents (such as fur, eyes, or contingent interaction with a human experimenter) some work has also argued that infants attribute physical *internal properties* to these objects as well [12,13]. In these experiments, 8-10-month-old infants show violation of expectancy effects when a self-propelled fur-covered entity is revealed to be hollow, that is, they apparently expected there to be something inside it. Setoh et al. (2013) [13] proposed that this could be a *biological* expectation, that is, infants expected there to be some internal biological features to this object that was self-propelled and had other agent-like qualities, and as a result looked longer when no such features were present. We presume, in the absence of any data that we have been able to find on the matter, that infants are seldom exposed to the insides of biological animate agents (indeed one might rather hope that they are not). Could they truly expect self-propelled objects with fur to have *biological* insides?

This proposal has substantial ramifications for our understanding of conceptual development. It suggests that, with little to no direct experience (we presume) with the insides of biological entities, they already have strong hypotheses about what such insides should look like. It argues that the idea of an "animate agent" with certain physical capabilities and mental states is actually a broader concept of "an animal", and that from early life we have extensive knowledge about the properties of animals. While it does not fully necessitate that knowledge of the specific category "animal" (as opposed to the broader category "animate agent") is innate, it suggests that it emerges very early and with a great deal of inferred information.

Research with preschoolers suggests that biological expectations are not entirely implausible. While children seldom see the insides of animals or complex machines, by age 4 they expect artifacts to have mechanical-looking insides and animals to have biological-looking insides [14]. That is not to say they have a precise sense of what the insides of a particular entity should look like. Rather, they have a sense that one *kind* of insides should go with one *kind* of entity. At around the same age, children have expectations that an object's insides should be related to its causal capabilities [15–17]. Thus, it is possible that infants' "insides" expectation is indeed a biological expectation [18–21].

However, Setoh et al. (2013) [13] also present an alternative proposal that this insides expectation is instead a reflection of general principles that self-propelled movement requires internal energy, and internal energy requires a source. Under this view, infants do not specifically expect *biological* insides, they just expect there to be *something* rather than *nothing*. In this proposal they do not necessarily understand animate agents as *animals* per se, but as an abstract category of entity that can generate its own motion and perhaps pursue goals. In other words, infants would also expect a robotic vacuum cleaner to have insides (particularly if they saw a human interacting with it in a social manner), but their insides expectation for the self-propelled vacuum cleaner might be, in its content, indistinguishable from their insides expectation for a pet cat. In fact, their entire understanding of the vacuum cleaner and the cat might be very similar. While infants are certainly sensitive to the difference between natural kinds and artifacts, even in plants [22], they may make no meaningful distinction between a biological animate agent and an artificial animate agent, at least early in life.

Current evidence does not distinguish between these two proposals. In part, this is because these different studies use different dependent variables and very different stimuli. In particular, Simons & Keil (1995) [14] were exceptional in studying *biological* insides specifically, and they exclusively used still images of animals that children could readily identify. In contrast, Li

et al. (in prep.) and Setoh et al. (2013) [12,13] used unfamiliar objects that had some features of animate agents, and presented videos to demonstrate whether or not they were self-propelled. Even in preschoolers, we do not know whether they understand novel self-propelled animate agents as biological in nature, or more abstract. More concretely, given a novel self-propelled object covered in fur, would preschoolers impute that it has biological insides, or identify it as a toy and expect it to have mechanical insides, or merely impute that there must be something inside it to explain its self-propelled movement while remaining agnostic as to what, specifically, those insides might be?

The current project seeks to test whether 4-year-old children 1) show a comparable pattern to infants in expecting self-propelled fur-covered objects to have insides more than inert objects, and if so 2) whether they expect the insides of a self-propelled fur-covered object to be specifically biological, mechanical, or show no preference. We also conducted a conceptual replication of Simons & Keil (1995) [14], testing whether these 4-year-olds do indeed expect mechanical objects to have mechanical insides and biological objects to have biological insides.

## Experiment

### Methods

All procedures, both in-person and online, were approved by the Rutgers University–Newark IRB under protocol 2020000399.

**Power analysis and preregistration.** Prior to data collection, we conducted a power analysis of what sample size would be required to reach 80% power to detect a moderate effect of test item (see below), defined as $w = .3$. This power analysis found that we would need 92 participants to safely achieve >80% power for our primary analysis of interest, so we set that as our target sample size. The study was preregistered at https://osf.io/n8pqm.

**Participants.** We ultimately collected data from 93 participants (mean age 54.8 months, range 48–63 months; 52 male, 40 female, 2 no gender reported), one of whom was excluded for refusing to answer any of the test items even after repeated prompting. 63 participants were run in preschool settings in the metropolitan Newark, NJ area (including the excluded participant), but due to the COVID-19 pandemic, 30 were run using online methods (see below).

**Stimuli.** The full script and all stimulus files can be found at https://osf.io/3xzad/. The critical stimuli consisted of the familiarization videos of Li et al. (in prep) Experiment 1. These videos each showed one puppet, an orange puppet covered in fur, or a blue puppet covered in feathers. One video, the "self-propelled" familiarization, showed one of the puppets moving back and forth across a stage apparently under its own power. Another video, the "inert" familiarization, showed a human actor standing behind the stage with their face concealed by the back of the stage, and reaching through a window to grab the other puppet as it sat in a pink tray. The puppet was then pushed and pulled back and forth across the stage with the same speed and timing by the human actor. The self-propelled familiarization included a short delay at the start to match the timing of the two videos as closely as possible. Which puppet was the self-propelled object and which was the inert object was counterbalanced between participants.

After each video, participants were shown three "X-rays" of the object in the video, which consisted of the puppet overlaid with an oval containing a monochrome image. An example test item is shown in Fig 1, for the orange fur-covered puppet. One "X-ray" showed mechanical-looking insides derived from a public-domain image of the clockworks of a church in Germany, while another showed biological-looking insides derived from a public-domain image of a torso MRI. A third "X-ray" showed only a gradient, indicating "empty" (participants learned that this represented "empty" during the training, see below). The "X-rays" were identical for all items, simply overlaid on the appropriate object.

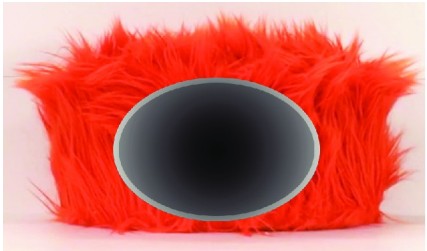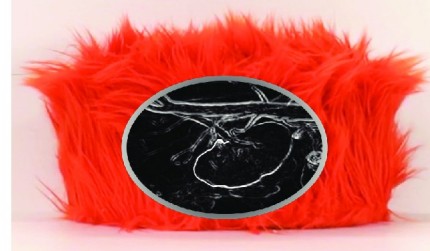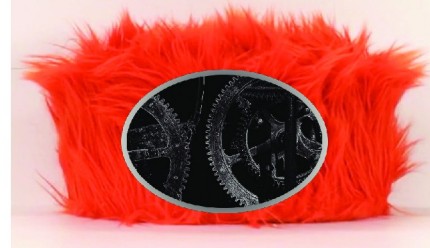

**Fig 1. Example of the "X-ray" images that children were asked to choose between.** The "Empty" insides are on the left, "Biological" in the middle, and "Mechanical" on the right.

We also created biological/mechanical intuition check items based on Simons & Keil (1995)(14). We used public-domain photographs of a sheep and motorcycle, and presented each photo to participants. Each one was then followed by a test item with three "X-rays" that were identical to the test items but rotated 90˚ (vertical rather than horizontal), overlaid on the image of the sheep and motorcycle. In addition, for training, we used a public-domain photo of a red bucket and overlaid it with an "empty" X-ray and a unique "sand" X-ray (see below).

**Procedure: In-person.** Participants were run in quiet spaces in their own pre-schools during school hours. The study was presented on a tablet device using the Qualtrics app [23]. The Qualtrics survey is available for download from the repository at https://osf.io/3xzad/.

The experiment began with a training item. Participants were asked if they knew what an X-ray was. Regardless of their response, all participants were provided with the following description: "An X-ray is a machine that can take a picture of what's inside something without opening it up. So if we take an X-ray of a bucket [shown image of red bucket] and it was empty, the X-ray shows us that it's empty [shown image of "empty" X-ray overlaid on red bucket], but then if we put sand in it, the X-ray shows us the sand [shown image of bucket filled with sand at the top, and a monochrome image of a "sand" texture overlaid on the body of the bucket]".

Participants were then shown an image of the two puppet-like objects side by side and told they were found on alien planets, that they had been sent to get X-rayed, and that the X-rays had gotten mixed up. They were told they would watch a video of each of the objects and then asked to tell us which X-ray went with each one.

Participants then completed two test trials. In each test trial, they first saw a familiarization video (either self-propelled or inert, as described above). The video was played in full, and the experimenter then advanced to the next page of the survey which showed the three X-rays for that item side-by-side in random order. Participants were asked "which X-ray shows what's inside this thing?" and could respond either by pointing or by tapping on the picture themselves.

After the test trials, participants completed two "biological/mechanical intuition" trials. In each trial they were shown either the image of the sheep or motorcycle and asked to say what it was (all participants were able to provide something roughly accurate, e.g., "sheep", "lamb"), and then shown the three X-rays for that item and asked which X-ray showed what was inside it, and they could respond the same way as the test trials.

**Procedure: Online.** Due to the COVID-19 pandemic, preschool data collection ended in early March, 2020. The study was subsequently modified to be run over a video call in order to comply with physical distancing guidelines. The study was modified in the following ways.

The approach was based on the procedure developed by TheChildLab.com [24]. For all items, participants were shown three options on different-colored backgrounds, and

responded by verbally naming the color behind their choice. To ensure children understood the answering protocol, an additional training trial was added to the very start of the experiment, in which participants were shown a triangle, circle, and square on the three different colors that would be used for all responses in the study, and asked to state which one was a circle using the color.

Participants were run in their own homes, with their parents present. (We were prepared to exclude participants for obvious parental interference, e.g., guiding the participants' choice, but there were no instances of overt parental interference in this sample.) We determined that screen-sharing in video-conferencing software presented video that was often low-resolution and at a very low frame rate, which would make the features of the objects in the test trial hard to distinguish (i.e., it might be hard to make out the fur/feathers) and make the motion of the objects jerky and unpredictable. The full experimental procedure was therefore re-created as a slides.com presentation (https://slides.com/). The experimenter has access to a presenter interface in their own browser which allows them to advance to the next slide, and the participants' browser advances accordingly.

Four presentations were created, one for each condition, and the parent was provided with a link to the appropriate presentation at the start of the video call. The participant watched the presentation in a browser while listening to the experimenter (as the video call was typically hidden behind the browser), while the experimenter controlled advancement through the slide-show on their own computer.

The registration was updated prior to the beginning of online data collection to reflect these changes (https://osf.io/pzkxw).

## Results

**Test items.** Children's responses to the test items can be found in Fig 2A, which shows the proportion of "mechanical insides", "biological insides", and "empty" responses for each item.

We had two primary questions of interest for our two test items: 1) whether preschoolers, like infants, would show a stronger expectation that there was *something* inside the self-propelled puppet compared to the inert puppet (i.e., whether they were less likely to choose the "empty" option for the self-propelled item), and 2) if they had intuitions about *what kind* of insides the puppet should have (i.e., comparing the proportion of biological to mechanical

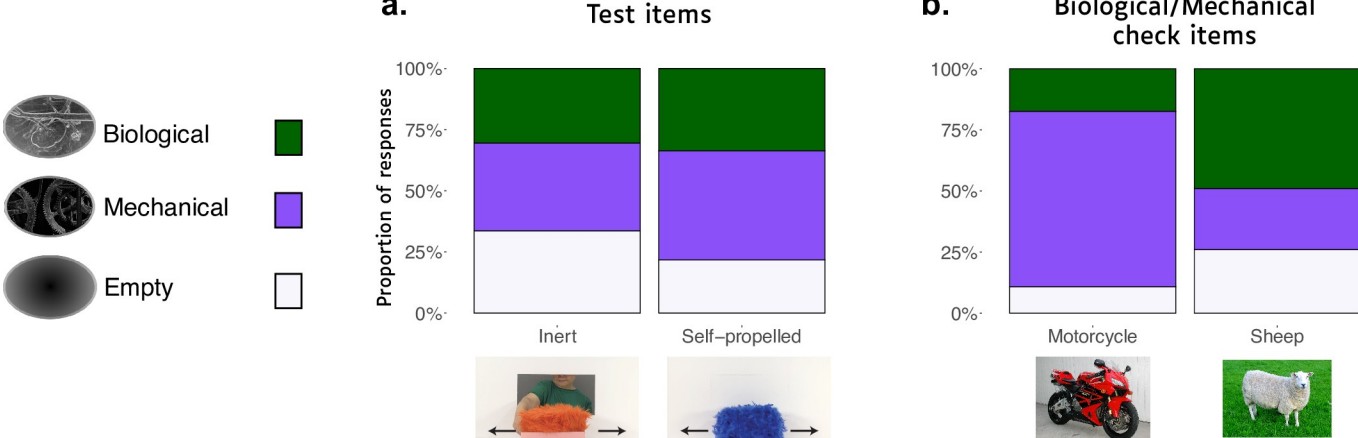

**Fig 2.** Proportion of "Empty", "Mechanical", and "Biological" responses for each item for the test items (left graph) and the biological/mechanical intuition check items (right graph).

responses both overall and for each item). Our preregistered analyses examined each of these in turn.

For question (1), we re-coded children's responses as either "empty" or "not empty", and conducted a mixed-model binary logistic regression to see whether the rate of "empty" responses varied by Item (self-propelled vs. inert). A chi-square goodness-of-fit analysis contrasting a model with Item as a predictor to a neutral model revealed that there was indeed a significant effect of Item, $\chi^2(1) = 3.998$, $p = .046$, $w = .21$. Participants were more likely to select the "empty" X-ray for the inert item (31/92) than the self-propelled item (20/92). This pattern of results was qualitatively similar across both the online and in-person samples.

For question (2), we removed all "empty" responses and conducted a further binary logistic regression comparing the rate of "mechanical" and "biological" responses to each test item. A chi-square goodness-of-fit analysis contrasting a model with Item as a predictor to a neutral model found no effect of item, $\chi^2(1) = .11$, $p = .74$. To determine if children had an overall preference for the "biological" or "mechanical" insides, we conducted a binomial exact test, which found no evidence that children preferred one or the other (59 "biological" and 74 "mechanical" responses, collapsing across both test items), $p = .22$.

In short, while 4-year-olds do share infants' intuition that a self-propelled fur-covered entity should have something inside it, they do not seem to have a specific expectation about *what* should be inside it, and show no evidence of a *biological* expectation in particular.

**Biological/Mechanical intuition check.** Responses to the intuition check items can be found in Fig 2B. We first conducted a preregistered analysis modeled on that used by Simons & Keil (1995) [14] to determine if we replicated their finding that 4-year-olds are better than chance at assigning biological insides to animals and mechanical insides to artifacts.

We recoded responses as either "accurate" or "inaccurate", yielding a score for each child between 0 and 2. We then conducted a two-tailed single-sample *t*-test comparing the average score against 1, which is a highly conservative definition of "chance responding" in this case (since they had three options for each item, actual random responding would generate a mean score of .667), but matches the analysis conducted by Simons and Keil (1995) [14]. Our participants performed significantly better than chance at identifying the appropriate insides for these items ($M = 1.21$, $SD = .75$), $t(91) = 2.64$, $p = .0096$, $d = .28$. This replicates the results of Simons and Keil, who also found that 4-year-old performed better than chance.

However, Simons & Keil (1995) [14] did not offer children an "empty" option, and there is additional insight to be gained by examining the full pattern of responses to these items. Therefore, we conducted a post-hoc analysis that was similar to the one we conducted for the test items, comparing the rate of "empty" responses to each item. This analysis found a significant effect of item, $\chi^2(1) = 25.58$, $p < .001$, $w = .51$. Surprisingly, participants were more likely to choose the "empty" option for the sheep (24/92) than the motorcycle (10/92). This may indicate that preschoolers may have a weaker idea of what biological insides look like compared to mechanical insides. To establish whether they preferred biological insides for the sheep at all, we also conducted a post-hoc test exact binomial test of the proportion of "biological" responses to the sheep item against chance (.33), which showed that they did select the biological option more often than chance overall (45/92), $p = .003$. While they may have chosen "empty" more often for the sheep item than the motorcycle, nonetheless they were consistent in selecting the biological insides over the mechanical or empty insides.

## General discussion

Preschool-aged children, like infants, expect self-propelled fur-covered objects to have insides. However, we find no evidence that they have specifically *biological* expectations. At the same

time, our participants succeeded at matching biological insides to an image of a real-world animal and mechanical insides to an image of a real-world artifact, replicating and extending past work (14) and indicating that they can match biological insides to entities that they identify as biological.

This experiment was not conducted with infants, and cannot directly assess Setoh et al. (2013)'s [13] proposal that 8-month-old infants have biological expectations about animate agents. However, it does provide evidence for an alternative view: Infants and children seek out internal causes for self-propelled motion from novel objects, but have only the most general expectations about what those causes are. This fits into a broader set of evidence showing that infants and children expect internal features to be causally important for objects that have features of animate agents. For example, infants track the identity of animate agents on the basis of internal rather than external properties when they are pitted against each [25,26], and as previously mentioned, preschoolers intuitively expect connections between internal properties and causal powers [15–17].

Even though preschoolers distinguish biological and mechanical insides [14], when it comes to causal connections between insides and observed behavior or properties, their intuitions may be much more abstract [27]. When there is no available external cause, children may readily infer that there must be an internal cause without knowing anything about the nature of that cause. Even when they have intuitions about what is biological versus mechanical, those intuitions are not specific. Children may not expect a specific set of causal insides (i.e., a specific set of identifiable parts), just some type of insides that could suffice as a causal explanation in principle.

An alternative explanation for children's responses is that preschoolers have more experience with fur-covered mechanical toys, and therefore were uncertain about whether the objects used in this study were animals or machines. It is clear from the biological/mechanical intuition check that if the majority of children believed that these objects were mechanical, they could have said so. Participants overwhelmingly chose the mechanical insides for the motorcycle, a familiar (and partially self-propelling) mechanical entity. Similarly, it is clear that if they believed these objects were animals, they would have overwhelmingly rejected the mechanical insides (if not favored the biological insides), which they also did not do. However, it is possible that children had different ideas about what the objects were based on idiosyncratic experiences. While the objects had fur and feathers, features which previous studies have found infants use as a cue to animate agency [13,27], their colors were thoroughly artificial (bright orange and royal blue). Children who recognized that such colors do not occur in nature may have inferred that the objects are toys, while those that did not may have believed they were animals. We might therefore expect this pattern of responses to shift somewhat with experiences. Adults, at least those from heavily industrialized cultures, might very well favor mechanical insides for the self-propelled object in these stimuli, but such a finding would have limited usefulness in understanding early-developing intuitions about animate agents.

Another, related concern is that some children might have (correctly) identified these objects as puppets, i.e., inferred that there was another agent (or part of an agent) inside them causing them to move. Children in this study were limited in what responses they could choose, and the presence of a hand or smaller agent was not one that we provided. Indeed, one reason we might have seen such 50/50 responding is that children did have a specific intuition about what is inside these agents, but it was not one of the options we presented. Future work could explore these possibilities by using a two-step task in which children are asked first if they think there is anything inside the object, and then given several options including mechanical, biological, a substance without causal power (e.g., sand), or even the presence of a smaller agent hiding under the object (i.e., showing that it is truly a puppet).

There is also a yet more abstract intuition that could explain these results: our findings show that preschoolers are less likely to pick the 'empty' insides for a self-propelled object compared to one that is moved by a human actor. The self-propelled object is demonstrating more complex behavior than its inert counterpart. Previous work has found that children at this age expect artifacts with more complex behavior to have more complex insides [16]. That heuristic, which is not specific to animate agents, could also be applied in this case of potentially animate agents. The 'empty' insides are simpler than both the biological and mechanical insides, and so children may simply have matched the more complex behavior to the more complex insides.

The matched-complexity explanation does not apply readily to the infant results, however. When infants see a self-propelled box that has no fur or other features of animate agents, they do not show any violation of expectancy when the box is revealed to be hollow [12] unless they also have evidence that a human interacts with it socially [13]. Similarly, 13-month-old infants do not track the identity of objects based on internal properties if they have no other cues to animate agency [26]. Infants clearly understand that the internal properties of animate agents are uniquely causally important, and that understanding does seem to rely in part on features that are derived from biology (fur, eyes) in addition to behavior. Yet, it is unclear whether these features are important because they are *biological*, or because they can be found on other animate agents.

Finally, it is worth considering how children responded to the *inert* object as well. Notably, while they picked the 'empty' response more often than in the self-propelled condition, they did not favor the 'empty' response over either the biological or mechanical insides. This is not unexpected. If the behavior of an object can be explained by an external cause, it provides no information about that object's properties whatsoever. Rather than assuming that the inert object was empty because it displayed no causal powers, children's responses seem to be random, which is what we might expect if the behavior they saw was uninformative. Of course, it is difficult to draw any conclusions from essentially random responding, but our findings raise the question of what behavior would *positively* indicate that an object was empty.

This project opens several avenues of important future work with infants, preschoolers, and even adults. With infants, there are critical questions about the role of these fur-like coverings. For example, future work could examine whether infants would have insides expectations about a self-propelled puppet covered in leaves or other features of plants, which infants and children clearly identify as biological [22], but not animate [28]. This would inform whether the critical aspect of fur is that it is simply a feature of natural kinds, or specifically a feature of familiar animate agents. With preschoolers, one could alter these puppets to be more or less animal-like. Intuitively, we would expect that the more the puppet has the features of familiar animals (e.g., eyes, limbs), the more children should treat it like a familiar animal and report it having biological insides.

It would also be worth investigating whether 4-year-olds, like infants, respond to other cues to agency, such as contingent social responding. Infants expect a box that beeps in response to a human experimenter and demonstrates self-propelled motion to have something inside it, even if it does not have fur [13]. This finding illustrates two notable points about the identification of agents. First, there is an impression, not yet carefully tested, that agent identification relies on adding together different kinds of cues, like social responding, self-propelled motion, fur, eyes [25,29,30], changing shape or size [7], and possibly indicators of goal-directedness like changing orientation to track a target [1]. Second, there may be no single *necessary* indicator of agency. Rather, the results we find here may be achievable with any sufficient combination of agentive traits. More work is needed to systematically vary different combinations of these cues to investigate this hypothesis in both infants and children, if not adults.

There is also an urgent need for more work on children's intuitions about the insides of familiar things. To our knowledge, this is only the second study to explicitly ask children about the insides of different causal systems on the dimension of natural kind or artifact (the first being Simons & Keil, 1995) [14]. In our study, the only time preschoolers selected the biological insides at above-chance rates were when we showed them a static picture of a familiar animal. The fact that we showed them a static picture is somewhat notable, when the animal is familiar they do not need direct evidence that it is self-propelled. However, while they picked the biological insides at above-chance rates for the image of the sheep, they did not have an overwhelming preference for the biological insides, and frequently selected the "empty" option. The role of seeing self-propelled movement deserves further investigation: did children frequently select the empty option because of a lack of experience with the insides of biological things, or because we only showed them a static image?

There are several other pertinent questions that have yet to be asked about the role of children's individual experience. Within this study, we have already made the proposal that biological attributions might increase the more animal-like the puppet appears to be. However, it would also be well worth investigating the role of industrialization in the child's everyday environment, and exposure to mechanical self-propelled entities more generally, in children's intuitions. There are also broader questions about how much children are exposed to information about insides directly. Are children matching on the basis of ontological category, that is, natural-looking insides with natural kinds and artifact-looking insides with artifacts, or do they have some exposure through illustrations or even live experience with the insides of mechanical things? These experiences likely vary substantially by individual, but perhaps the most interesting question is what kind of insides expectations are found in children with *no* such experience, which would give greater insight into intuitions derived from more general principles about the world.

More generally, future work has the opportunity to use these insides expectations, however nonspecific they may be, to delve into infants' and children's understanding of animate agency, the natural world, and the relationship between the two.

## Acknowledgments

The authors would like to thank Yiping Li for the initial creation of the puppet stimuli, and Micaela Generali, Marina Hanna, and Sierra Guequierre for assistance with data collection. Thanks also to members of the Computational Cognitive Development lab for helpful feedback on an earlier draft of this manuscript.

## Author Contributions

**Conceptualization:** Jonathan F. Kominsky, Patrick Shafto, Elizabeth Bonawitz.

**Data curation:** Jonathan F. Kominsky.

**Formal analysis:** Jonathan F. Kominsky.

**Funding acquisition:** Patrick Shafto, Elizabeth Bonawitz.

**Investigation:** Jonathan F. Kominsky.

**Methodology:** Jonathan F. Kominsky.

**Project administration:** Jonathan F. Kominsky.

**Resources:** Patrick Shafto, Elizabeth Bonawitz.

**Supervision:** Patrick Shafto, Elizabeth Bonawitz.

**Visualization:** Jonathan F. Kominsky.

**Writing – original draft:** Jonathan F. Kominsky.

**Writing – review & editing:** Jonathan F. Kominsky, Patrick Shafto, Elizabeth Bonawitz.

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
