## [Decision Letter · Decision Letter 0]

23 Dec 2020

PONE-D-20-34496

"There's something inside": Children's intuitions about animate agents

PLOS ONE

Dear Dr. Kominsky,

Thank you for submitting your manuscript "There's something inside": Children's intuitions about animate agents” for review and consideration for publication in PLOS ONE. I have now received 3 reviews of the manuscript, and I have carefully read the manuscript myself. The reviews are appended to this email.

The Reviewers and I agree that you are addressing an interesting and important question about pre-schoolers’ predictions regarding the insides of self-propelled objects. At the same time, they all raised some concerns about the conclusions you drew based on the present data. Specifically, the main issue, which was mentioned by both Reviewer 1 and Reviewer 2 is that the stimuli you were using for the main experiment could be well considered by the children as mechanical toys, rather than animals. Therefore, considering that at this age probably they already have some experience with self-propelled toys, their expectations regarding the inside of the present objects may not be informative about their biological understanding of animals. I agree with this concern of the reviewers and I am afraid that indeed this could be a serious limitation of your study. Therefore I think unless you can provide sufficient evidence that the subjects you tested perceived the fur-covered self-propelled objects as real animals, you should reformulate the conclusions of your study.

As you will see, to deal with this concern, Reviewer 1 proposed to run the same experiment with adult subjects, whereas Reviewer 2 suggested to use stimuli of more realistic animate objects. While I think both of these suggested additional experiments could be informative in relation to your study, I am afraid that they still cannot guarantee that you could hold the same conclusions you have now. Therefore, in light of all these issues, while I am inviting you to submit a revised version of your paper, I would suggest you to prepare your revision after reconsidering your main hypothesis and the potential conclusions you would be able to make based on the present experiment (and on any additional experiments you might plan to run). Also, in case you decide to resubmit a revised version of your paper, please address also all the additional comments of the reviewers.

Please submit your revised manuscript in 60 days. If you will need more time than this to complete your revisions, please reply to this message or contact the journal office at plosone@plos.org. Please include the following items when submitting your revised manuscript:

We look forward to receiving your revised manuscript.

Kind regards,

Hanna Marno

Academic Editor

PLOS ONE

Journal Requirements:

2. Our internal editors have looked over your manuscript and determined that it is within the scope of our Cognitive Developmental Psychology Call for Papers. The Collection will encompass a diverse range of research articles in developmental psychology, including early cognitive development, language development, atypical development, cognitive processing across the lifespan, among others, with an emphasis on transparent and reproducible reporting practices.  Additional information can be found on our announcement page: https://collections.plos.org/s/cognitive-psychology.  If you would like your manuscript to be considered for this collection, please let us know in your cover letter and we will ensure that your paper is treated as if you were responding to this call. Please note that being considered for the Collection does not require an additional peer review beyond the journal’s standard process and will not delay the publication of your manuscript if it is accepted by PLOS ONE. If you would prefer to remove your manuscript from collection consideration, please specify this in the cover letter.

Reviewers' comments:

Reviewer's Responses to Questions

**Comments to the Author**

1. Is the manuscript technically sound, and do the data support the conclusions?

Reviewer #1: Partly

Reviewer #2: Partly

Reviewer #3: Partly

2. Has the statistical analysis been performed appropriately and rigorously? 

Reviewer #1: Yes

Reviewer #2: Yes

Reviewer #3: Yes

3. Have the authors made all data underlying the findings in their manuscript fully available?

Reviewer #1: Yes

Reviewer #2: Yes

Reviewer #3: Yes

4. Is the manuscript presented in an intelligible fashion and written in standard English?

Reviewer #1: Yes

Reviewer #2: Yes

Reviewer #3: Yes

5. Review Comments to the Author

Reviewer #1: The authors examined whether preschoolers make systematic predictions about the insides of self-propelled objects. The paper is concise and clear and offers an interesting method for bridging two related lines of research that could not be directly compared before.

One comment relevant to present paper:

• It would be helpful to have adult data. The paper implies that the mature response pattern is systematic selection of the biological insides. This is implied by the developmental story wherein young children have ‘abstract’ expectations about insides of self-propelled furry objects but then acquire more specific biological understandings with age. But the self-propelled object looks artificial and engaged in simplistic movements that imply a toy rather than an animal. If adults systematically predict mechanical insides then that seriously limits the interpretation of the study. In particular, it may be that the developmental trajectory with these stimuli is a biological to mechanical switch: Infants make biological attributions, adults make mechanical attributions, and 4-year-olds are in the early-states of that switch. Though this limitation could also be present in the infant work, it is also highly plausible that infants lack requisite knowledge about which kinds of colors, materials (fur/feathers), and movements are commonly artificially reproduced. So, it is possible that infants are making biological attributions. In lieu of adult participants, it would be helpful if this possibility was addressed in the General Discussion.

A few thoughts for authors to consider for future directions:

• Authors might consider a separate project investigating exactly this question of animal / artifact attributions; It could be really interesting to see how expectations about the appearance and locomotion/behavior of animals develops. This would be particularly interesting cross-culturally. You may find that children continue to perceive the self-propelled object as an animal if they lack exposure to artifacts with bright colors and the ability to move (i.e., children’s toys in industrialized contexts). Whereas children with exposure to these kinds of artificial objects might begin to make mechanical inferences.

• In a future study, authors may consider a two-step procedure: First ask children whether they think the object is empty or not, and then ask what they think the insides are. For the ‘Inert’ condition: Presently, if a third of children expect no insides and two-thirds expect insides (but are 50/50 between biological and mechanical), their responses could resemble chance-responding overall when they are systematically expecting insides. For the ‘Self-propelled’ condition: Children that predict insides strongly (~80% of the time) but are 50/50 between biological and mechanical look only weakly different from chance when they might be very different from chance: My guess is you would find children selecting ‘has insides’ 50% of the time in the ‘Inert’ condition and 80% of the time in the ‘Self-propelled’ condition, which would better reflect the strength of their intuitions rather than the current results which look weaker that I bet children’s expectations really are.

• For future work, it may be worth exploring the addition of psychological behaviors, such as interacting contingently or pursuing goals.

Reviewer #2: The manuscript addresses an important issue in conceptual development concerning how children represent the insides of animate and inanimate objects. The authors clearly identify a gap in the literature and ask whether children possess concrete biological expectations about the animals’ insides. The study replicated previous findings with infants and preschool-age children and the authors conclude that 4-year-olds have abstract expectations about the insides of self-propelled objects, but don’t distinguish between mechanical and biological insides. While I acknowledge that this is an interesting study, I have some major concerns that I mention below:

1. Setoh, et. al (2013) and others (Taborda-Osorio & Cheries, 2018) have proposed that infants infer the presence of internal material properties whenever an object exhibits two features at once: it is self-propelled and agentive. “Agentivity” refers to those features in the object’s behavior that are diagnostic of intentionality; such as, contingent behavior, eyes or fur. In the introduction section this distinction it’s unclear and the authors seem to imply more than once that the key diagnostic feature is only the self-propelled motion. Accordingly, they try to replicate one of the experiments of the Setoh, et. al.’ study without giving much explanation as to why the presence of both self-propelled behavior and fur covering the object would allow children in principle to infer internal material properties. I think this issue deserves further elaboration in both the introduction and the discussion section.

2. The study compares the results of two tests, the self-propelled and the biological/mechanical intuition test. I find the results of both tests contradictory, in both cases children are supposed to represent an object as an “animal”, according to the results in the first test they don’t have biological expectations but in the second one they do have biological expectations, replicating the original findings. Why? The authors stress that in one test the objects are still while in the other are self-propelled, but again if both entities are represented like animals, we would expect a similar pattern of results. This issue should be brought out in the discussion section. For instance, are the insides of self-propelled objects less clearly identify with something biological?

3. Related to the previous point, one possibility is that the children in the current study are not representing the self-propelled object like an animal like infants do. Unlike 8-month-old infants, 4-year-olds have a great deal of knowledge about how a typical animal looks like and they also have experience with mechanical toys. A self-propelled fur-covered object may be something rather atypical as an animal exemplar, in between a mechanical toy and a real animal (also notice that mechanical fur-covered toys are pretty common today). Something striking in the pattern of results is that in the self-propelled condition children have a (significant?) preference for mechanical insides compared to empty insides, but probably a non-significant difference between empty and biological insides. Thus, one possibility is that the children here represent the object like a mechanical toy and are biased to attribute mechanical insides which could be more coherent with their daily-life experience.

4. I believe the hypothesis for biological expectations is ruled out too soon. What would happen if children witness a more realistic animate object (contingent behavior, eyes?)? would they still be agnostic as to what type of insides they have? If under these conditions they still don’t show any preference for one or the other I think the conclusion would be better supported. I believe the study would benefit a great deal from a second experiment with stimuli displaying more agentive features. I insist that the infants and children’s inference about the presence of insides is a matter of both self-propelled motion and also agentive features (see also Opfer & Siegler, 2004). Stronger evidence for agentive behavior may lead children to prefer the biological insides in the test. If this is the case, the comparison across both experiments would be very informative. In its current form, the study is mainly about children’s intuitions about the insides of self-propelled objects rather than the insides of animate agents.

Reviewer #3: In "There's something inside": Children's intuitions about animate agents, the authors tested whether 4-year-olds have intuitions about whether or not self-propelling vs inert objects have insides, and whether they have intuitions about the specifics of the insides. They tested this by showing novel objects (fur-covered boxes) that were either self-propelling or inert, and asked participants to choose between “X-rays” of the insides (empty, mechanical, or biological). The 4-year-olds were more likely to say that self-propelled objects are non-empty, but did not have an intuitions regarding the specifics of the insides of unknown objects.

The paper is well-written and a pleasure to read. The introduction clearly lays out the current knowledge and the question at hand, the design is straight-forward, and the authors have done a great job at moving the study online to test approximately 1/3 of the remaining participants after the pandemic hit. The authors have pre-registered their study, conducted a power analysis beforehand to determine the number of participants, and made their materials available, all of which is appreciated. I do still have a few questions and comments.

Main comments:

My first comment pertains to the results of RQ1. Although you did a power analysis and found a statistically significant difference (p=.046), I wonder whether the result is as strong as the discussion section would make us believe. Due to publication bias, the estimated effect size used for the power analysis is often larger than the true effect size. I would be curious to know whether the authors have calculated the true effect size for this finding and whether that effect size reasonably warrants their conclusions? The authors’ claims are quite strong for a result that does not seem to have a convincing large effect size. Moreover, I would encourage the authors to calculate the Bayes Factor to determine whether there truly is more evidence for the alternative hypothesis than for the null hypothesis. A Bayesian chi-square analysis can be conducted in JASP, and the authors could even use the responses from the inert condition as the null hypothesis. My worries on this matter stem from the ideas regarding the uncanny p-mountain (see http://www.the100.ci/2018/02/15/the-uncanny-mountain-p-values-between-01-and-10-are-still-a-problem/) and the relatively low (only a third) “empty”-responses for inert objects.

My second comment concerns the conclusions and discussion section. Although there is some previous literature that indicates that children may have causal beliefs about the insides of an objects and its ability to self-propel, these beliefs were not tested in this study. I think this warrants a more nuanced discussion differentiating between about what can be concluded directly from the study itself and what potential implications may be considered given previous findings. I recognise that the authors have already nuanced their conclusions in part, but I feel there is still too much of a jump from the measurements of their design to the way the results are discussed. In addition to a clearer distinction, a few sentences on how the parameters (design, measures, age groups, etc.) of the previous studies (that found causal beliefs relating to insides and agency) relates to the authors’ study might help the reader to better understand in which ways the earlier findings can be generalised to this study and where potential differences might be.

Minor comments:

How did you deal with potential parental interference during the online version of the study?

The data on OSF is only available after requesting access (no problems accessing the pre-registration). In the end I managed to find a way to the data via the pre-registration, but you might want to have another look at the settings or mention the requirement for requesting access.

References: It is difficult to see that (14) refers to the same paper as Simons & Keil (1995) without having to scroll back and forth. Obviously, this is due to the referencing system of the journal and not the authors’ doing. However, maybe the authors can do something to make it easier on the reader (e.g. write such situations as: Simons and Keil (1995) were exceptional [...] could readily identify (14).)? I don’t know whether PLOS ONE allows the authors to do this and it is only a very minor point.

6. PLOS authors have the option to publish the peer review history of their article (what does this mean?). If published, this will include your full peer review and any attached files.

Reviewer #1: No

Reviewer #2: No

Reviewer #3: No

---

## [Author Response · Author response to Decision Letter 0]

23 Feb 2021

Specific responses to reviewers and the editor are included in the cover letter as part of the submission.

---

## [Decision Letter · Decision Letter 1]

6 Apr 2021

PONE-D-20-34496R1

"There's something inside": Children's intuitions about animate agents

PLOS ONE

Dear Dr. Kominsky,

Thank you for submitting your revised manuscript "There's something inside": Children's intuitions about animate agents" for review and consideration for publication in PLOS One.

We were fortunate to receive another set of comments from three of the original Reviewers. All of them found your paper to be very responsive and think that this work makes an important contribution to the field. I agree with this assessment. Reviewer 3 also had some remaining suggestions for a minor revision, including a suggestion to report your observed effect sizes and also to discuss the number of “empty” responses related to RQ1. I would like to invite you to revise your paper one more time in response to these suggestions.

Please submit your revised manuscript in 30 days. If you will need more time than this to complete your revisions, please reply to this message or contact the journal office at plosone@plos.org. Please include the following items when submitting your revised manuscript:

We look forward to receiving your revised manuscript.

Kind regards,

Hanna Marno

Academic Editor

PLOS ONE

Journal Requirements:

Reviewers' comments:

Reviewer's Responses to Questions

**Comments to the Author**

1. If the authors have adequately addressed your comments raised in a previous round of review and you feel that this manuscript is now acceptable for publication, you may indicate that here to bypass the “Comments to the Author” section, enter your conflict of interest statement in the “Confidential to Editor” section, and submit your "Accept" recommendation.

Reviewer #1: All comments have been addressed

Reviewer #2: All comments have been addressed

Reviewer #3: (No Response)

2. Is the manuscript technically sound, and do the data support the conclusions?

Reviewer #1: Partly

Reviewer #2: Yes

Reviewer #3: Yes

3. Has the statistical analysis been performed appropriately and rigorously? 

Reviewer #1: Yes

Reviewer #2: Yes

Reviewer #3: Yes

4. Have the authors made all data underlying the findings in their manuscript fully available?

Reviewer #1: Yes

Reviewer #2: Yes

Reviewer #3: Yes

5. Is the manuscript presented in an intelligible fashion and written in standard English?

Reviewer #1: Yes

Reviewer #2: Yes

Reviewer #3: Yes

6. Review Comments to the Author

Reviewer #1: Concerns about the study design limit what we can learn from the data. But, I think the paper offers an interesting theoretical perspective and methodological approach that can generate future research, and so should be communicated sooner rather than later. I think the authors adequately addressed the limitations of the study in the general discussion in this revision, and so that is sufficient to ensure the findings are interpreted appropriately by readers.

Reviewer #2: (No Response)

Reviewer #3: I would like to thank the authors for the substantial work they did to revise discussion of their findings. In my opinion, it reflects the findings better and improved the paper a lot.

I have two remaining comments:

Although I take the point about preregistration as a means of showing that the analysis was planned and not a lucky shot, I do not think that preregistering their power analysis (based on previous literature) necessarily means it is unaffected by publication biases. It is not unlikely that the calculated sample size is based on an overestimated effect size. I suggest the authors report the observed effect sizes to provide better estimates for future studies to calculate power.

Regarding the discussion, much of it is focused on biological vs mechanical and it offers an appropriately nuanced view now. However, I think it would be good if the authors also spend a few words in the discussion on the number of “empty” responses related to RQ1. Although the difference with self-propelled was significant, the number of empty responses for inert items was only 31/92. What consequences does it have for the validity of the research question if children are not very likely to consider any object empty? What does it mean for agency assessment that in 61 cases, the child attributed insides to an inert object as well?

7. PLOS authors have the option to publish the peer review history of their article (what does this mean?). If published, this will include your full peer review and any attached files.

Reviewer #1: No

Reviewer #2: No

Reviewer #3: No

---

## [Author Response · Author response to Decision Letter 1]

13 Apr 2021

We are appreciative of the chance to have revised the manuscript and glad that the reviewers were generally satisfied by our previous set of revisions. Reviewer 3 has raised two final comments which we address in the cover letter and revised manuscript. 1. R3 requested we report effect sizes, which we now do for all significant results. 2. R3 raised an interesting question about interpreting the 'empty' responses. We have added a paragraph to the general discussion (p. 15) addressing this issue. See the cover letter for more details.

---

## [Decision Letter · Decision Letter 2]

20 Apr 2021

"There's something inside": Children's intuitions about animate agents

PONE-D-20-34496R2

Dear Dr. Kominsky,

Thank you for submitting the revised manuscript "There's something inside": Children's intuitions about animate agents (PONE-D-20-34496R2) and addressing the last comments of Reviewer 3. I am pleased to inform you that your manuscript has been judged scientifically suitable for publication and will be formally accepted for publication once it meets all outstanding technical requirements.

Kind regards,

Hanna Marno

Academic Editor

PLOS ONE

Additional Editor Comments (optional):

Reviewers' comments:

Reviewer's Responses to Questions

**Comments to the Author**

1. If the authors have adequately addressed your comments raised in a previous round of review and you feel that this manuscript is now acceptable for publication, you may indicate that here to bypass the “Comments to the Author” section, enter your conflict of interest statement in the “Confidential to Editor” section, and submit your "Accept" recommendation.

Reviewer #3: All comments have been addressed

2. Is the manuscript technically sound, and do the data support the conclusions?

Reviewer #3: (No Response)

3. Has the statistical analysis been performed appropriately and rigorously? 

Reviewer #3: (No Response)

4. Have the authors made all data underlying the findings in their manuscript fully available?

Reviewer #3: (No Response)

5. Is the manuscript presented in an intelligible fashion and written in standard English?

Reviewer #3: (No Response)

6. Review Comments to the Author

Reviewer #3: (No Response)

7. PLOS authors have the option to publish the peer review history of their article (what does this mean?). If published, this will include your full peer review and any attached files.

Reviewer #3: No

---

## [Editor Report · Acceptance letter]

22 Apr 2021

PONE-D-20-34496R2 

“There’s something inside”: Children’s intuitions about animate agents 

Dear Dr. Kominsky:

I'm pleased to inform you that your manuscript has been deemed suitable for publication in PLOS ONE. Congratulations! Your manuscript is now with our production department. 

Kind regards, 

on behalf of

Dr. Hanna Marno 

Academic Editor

PLOS ONE